# First-Trimester Screening for Fetal Growth Restriction and Small-for-Gestational-Age Pregnancies without Preeclampsia Using Cardiovascular Disease-Associated MicroRNA Biomarkers

**DOI:** 10.3390/biomedicines10030718

**Published:** 2022-03-19

**Authors:** Ilona Hromadnikova, Katerina Kotlabova, Ladislav Krofta

**Affiliations:** 1Department of Molecular Biology and Cell Pathology, Third Faculty of Medicine, Charles University, 100 00 Prague, Czech Republic; katerina.kotlabova@lf3.cuni.cz; 2Institute for the Care of the Mother and Child, Third Faculty of Medicine, Charles University, 147 00 Prague, Czech Republic; ladislav.krofta@upmd.eu

**Keywords:** cardiovascular microRNAs, early pregnancy, fetal growth restriction, gene expression, prediction, screening, small for gestational age, whole peripheral venous blood

## Abstract

The goal of the study was to determine the early diagnostical potential of cardiovascular disease-associated microRNAs for prediction of small-for-gestational-age (SGA) and fetal growth restriction (FGR) without preeclampsia (PE). The whole peripheral venous blood samples were collected within 10 to 13 weeks of gestation from singleton Caucasian pregnancies within the period November 2012 to March 2020. The case-control retrospective study, nested in a cohort, involved all pregnancies diagnosed with SGA (*n* = 37) or FGR (*n* = 82) without PE and 80 appropriate-for-gestational age (AGA) pregnancies selected with regard to equality of sample storage time. Gene expression of 29 cardiovascular disease-associated microRNAs was assessed using real-time RT-PCR. Upregulation of miR-16-5p, miR-20a-5p, miR-146a-5p, miR-155-5p, miR-181a-5p, and miR-195-5p was observed in SGA or FGR pregnancies at 10.0% false positive rate (FPR). Upregulation of miR-1-3p, miR-20b-5p, miR-126-3p, miR-130b-3p, and miR-499a-5p was observed in SGA pregnancies only at 10.0% FPR. Upregulation of miR-145-5p, miR-342-3p, and miR-574-3p was detected in FGR pregnancies at 10.0% FPR. The combination of four microRNA biomarkers (miR-1-3p, miR-20a-5p, miR-146a-5p, and miR-181a-5p) was able to identify 75.68% SGA pregnancies at 10.0% FPR in early stages of gestation. The detection rate of SGA pregnancies without PE increased 4.67-fold (75.68% vs. 16.22%) when compared with the routine first-trimester screening for PE and/or FGR based on the criteria of the Fetal Medicine Foundation. The combination of seven microRNA biomarkers (miR-16-5p, miR-20a-5p, miR-145-5p, miR-146a-5p, miR-181a-5p, miR-342-3p, and miR-574-3p) was able to identify 42.68% FGR pregnancies at 10.0% FPR in early stages of gestation. The detection rate of FGR pregnancies without PE increased 1.52-fold (42.68% vs. 28.05%) when compared with the routine first-trimester screening for PE and/or FGR based on the criteria of the Fetal Medicine Foundation. Cardiovascular disease-associated microRNAs represent promising early biomarkers with very suitable predictive potential for SGA or FGR without PE to be implemented into the routine screening programs.

## 1. Introduction

Small-for-gestational-age (SGA) fetuses are defined as those ones with the estimated fetal weight (EFW) or abdominal circumference (AC) below the 10th percentile for the evaluated gestational age [1,2,3,4]. Several studies have demonstrated an association between abnormal placental development and SGA [5,6,7,8,9].

The definitions of early-onset (before 32 weeks of gestation) and late-onset (after 32 weeks of gestation) fetal growth restriction (FGR) in the absence of congenital anomalies are based on international Delphi consensus [4,10]. Early FGR fetuses are defined as fetuses with AC/EFW below the 3rd percentile or absent end-diastolic flow in an umbilical artery or fetuses with AC/EFW below the 10th percentile combined with uterine artery pulsatility index (UtA-PI) above the 95th percentile and/or umbilical artery pulsatility index (UA-PI) above the 95th percentile. Late FGR fetuses are defined as fetuses with AC/EFW below the 3rd percentile or those ones with the presence of at least 2 out of 3 of the following criteria: 1. AC/EFW below the 10th percentile, 2. AC/EFW crossing centiles > 2 quartiles on growth centiles, 3. cerebroplacental ratio (CPR) below the 5th percentile or UA-PI above the 95th percentile [4,10].

Nowadays, the early risk estimation model (between 11 and 14 weeks of gestation) for trisomy 21, 18, and 13, preeclampsia (PE), FGR, and preterm delivery proposed by the Fetal Medicine Foundation (FMF) has usually been used worldwide [11]. This risk estimation model is founded on the combination of maternal history, maternal clinical characteristics (body mass index (BMI), and mean arterial blood pressure (MAP)), maternal serum biomarkers (pregnancy-associated plasma protein A (PAPP-A), and placental growth factor (PIGF)), and mean UtA-PI [12,13,14,15]. In addition, the combination of maternal factors, MAP, PIGF, and UtA-PI used to screen pregnancies for preterm PE was simultaneously used to predict the occurrence of SGA. This approach identified at 10% false positive rate (FPR) in total 45.8% of SGA neonates born before 37 weeks of gestation and 56.3% of SGA neonates delivered before 32 weeks of gestation regardless of PE occurrence [10]. In the case of the onset of SGA without PE, the detection rate was a bit lower at 10% FPR (31.3% of SGA pregnancies delivering before 37 weeks of gestation and 34.9% of SGA pregnancies delivering before 32 weeks of gestation) [11].

The latest study demonstrated that the combination of all markers including maternal factors, MAP, UtA-PI, PIGF, and PAPP-A predicted at 10% FPR in total 48.6% of SGA neonates delivered before 37 weeks of gestation (AUC 0.795) and 59.1% of SGA neonates delivered before 32 weeks of gestation (AUC 0.8257) regardless of PE occurrence [16]. In the case of the onset of SGA without PE, the detection rates were again a bit lower: 44.4% of SGA pregnancies delivered before 37 weeks of gestation (AUC 0.7674) and 51.2% of SGA pregnancies delivering before 32 weeks of gestation (AUC 0.7751) [16].

Combined screening for FGR at 11 to 14 weeks of gestation, based on the combination of maternal characteristics, MAP, UtA-PI, PAPP-A, and PIGF, reached 67.2% sensitivity at 82.7% specificity (AUC 0.61, PPV 70.4 and NPV 79.1) [17].

According to the FMF algorithm, pregnancies at high risk of the onset of preterm PE are recommended to receive daily low-dose aspirin (ASA) from 11 to 14 weeks of gestation until 37 weeks of gestation to reduce the incidence of preterm PE [18,19]. Subsequently, prevention by the prophylactic use of ASA was also recommended in preterm SGA neonates identified by first-trimester screening for PE, since in the ASPRE trial, use of aspirin reduced the incidence of SGA in neonates delivered before 37 and 32 weeks of gestation [20].

Nevertheless, according to the guidelines of the American College of Obstetricians and Gynecologists (ACOG) and the Society for Maternal-Fetal Medicine [21], the use of prophylactic ASA for the prevention of recurrent FGR is not currently recommended in women without other risk factors for PE.

We focused on the exploration of gene expression profiles of selected cardiovascular disease-associated microRNAs in the whole peripheral venous blood of women during the early stages of gestation with the aim to assess the predictive potential for SGA or FGR without PE. In addition, we compared the predictive rate for FGR calculated on the base of microRNA gene expression profiles with the predictive rate for PE and/or FGR calculated by the Astraia Obstetrics program using the FMF risk algorithm [11]. Up to now, no reports on microRNA gene profiling in the whole peripheral venous blood in the early stages of gestation are at disposal in pregnancies with subsequent onset of SGA or FGR.

Intracellular and extracellular microRNA expression profiles can differ significantly for various reasons. Circulating biomarkers, detectable in maternal plasma, serum, or body fluids, usually reflect expression profiles in apoptotic bodies and extracellular vesicles derived from various types of cells present in the human body. On the other hand, whole peripheral blood leukocyte lysates usually reflect intracellular expression profile corresponding with actual systemic response of the human body to the current stage (normal or abnormal course of gestation).

## 2. Materials and Methods

### 2.1. Patients Cohort

Singleton pregnancies of Caucasian descent only were involved in the retrospective case-control study held within the period November 2012 to March 2020. The whole peripheral venous blood samples were collected during the first-trimester gestation at 10–13 weeks in the Institute for the Care of Mother and Child, Prague, Czech Republic. Finally, 4469 out of 6440 pregnancies had complete medical records since they were delivered in the Institute for the Care of Mother and Child, Prague, Czech Republic. Out of these 4469 pregnancies, 82 women were confirmed to have growth-restricted fetuses, and 37 women were diagnosed to carry small-for-gestational-age fetuses. In detail, 5 FGR pregnancies were delivered before 32 weeks of gestation (early FGR) and 77 pregnancies after 32 weeks of gestation (late FGR).

The control group included 80 women with a normal course of gestation that delivered healthy infants after completing 37 weeks of gestation with a weight above 2500 g. The control group was selected with regard to the uniformity of gestational age at sampling and storage times of biological samples.

The clinical characteristics of pregnancies are outlined in Table 1.

All participants signed informed consent. The Ethics Committee of the Institute for the Care of the Mother and Child and The Ethics Committee of the Third Faculty of Medicine, Charles University, approved the study (Implication of microRNAs in maternal circulation for diagnosis and prediction of placental insufficiency, date of approval: 7 April 2011). All procedures were in compliance with the Helsinki Declaration of 1975, as revised in 2000.

### 2.2. Prediction of the Occurrence of PE and FGR

Astraia Obstetrics Programme (Astraia – software for women´s health, obstetric and gynaecological database, application version 1.27.3; Astraia FMF – first trimester screening for trisomy 21, version 4.4, Germany) was used to calculate the risks for chromosomal aneuploidies (trisomy 21, 18, and 13), preeclampsia, fetal growth restriction, and preterm delivery in the first trimester of gestation using the FMF risk algorithm [11] developed in close collaboration with the Fetal Medicine Foundation.

### 2.3. Processing of Samples

Homogenized leukocyte lysates containing lymphocytes, monocytes, and granulocytes were prepared from 200 µL maternal whole peripheral venous blood samples immediately after collection using QIAamp RNA Blood Mini Kit (Qiagen, Hilden, Germany) according to manufacturer’s instructions Firstly, lysis of erythrocytes was performed using EL buffer and then pelleted leukocytes were stored in a mixture of RLT buffer and β-mercaptoethanol (β-ME) at −80 °C until further processing. β-mercaptoethanol eliminates ribonucleases released during cell lysis. Numerous disulfide bonds make ribonucleases very stable enzymes, so β-mercaptoethanol is used to reduce these disulfide bonds and irreversibly denature the proteins. This prevents them from digesting the RNA during its storage and subsequent extraction procedure. Homogenized cell lysates can be stored like this at −80 °C for several months. To keep the requirements of the manufacturer, we performed real-time RT-PCR analyses regularly every six months to process the collection of frozen samples derived from normal and complicated pregnancies.

Subsequently, a mirVana microRNA Isolation kit (Ambion, Austin, TX, USA) was used to isolate RNA fractions highly enriched for small RNAs from whole peripheral blood leukocyte lysates.

The concentration and quality of RNA were assessed using a NanoDrop ND-1000 spectrophotometer (NanoDrop Technologies, Wilmington, DE, USA). The A(260/280) absorbance ratio of isolated RNA samples was 1.8–2.1, demonstrating that RNA samples were pure and could be used for further analysis. The concentration of isolated RNA ranged within 2.00.0 ng/µL.

Gene expression of 29 cardiovascular disease-associated microRNAs (miR-1-3p, miR-16-5p, miR-17-5p, miR-20a-5p, miR-20b-5p, miR-21-5p, miR-23a-3p, miR-24-3p, miR-26a-5p, miR-29a-3p, miR-92a-3p, miR-100-5p, miR-103a-3p, miR-125b-5p, miR-126-3p, miR-130b-3p, miR-133a-3p, miR-143-3p, miR-145-5p, miR-146-5p, miR-155-5p, miR-181a-5p, miR-195-5p, miR-199a-5p, miR-210-3p, miR-221-3p, miR-342-3p, miR-499a-5p, and miR-574-3p) was determined.

mRNAs of appropriate microRNAs were reverse transcribed into cDNA by TaqMan MicroRNA Assays containing miRNA-specific stem loop primers and TaqMan MicroRNA Reverse Transcription Kit (Applied Biosystems, Branchburg, USA). The total reaction volume was 10 µL. Furthermore, 3 µL of cDNA were mixed with the components of TaqMan MicroRNA Assays containing specific primers and the TaqMan MGB probes and the components of the TaqMan Universal PCR Master Mix (Applied Biosystems, Branchburg, NJ, USA). The total reaction volume was 15 µL. Reverse transcription and real-time qPCR were performed on 7500 Real-Time PCR System using TaqMan PCR conditions set up in the TaqMan guidelines. Reverse transcription thermal cycling parameters were the following: 30 min at 16 °C, 30 min at 42 °C, 5 min at 85 °C, and then held at 4 °C. Real-time qPCR thermal cycling parameters were the following: 2 min at 50 °C, 10 min at 95 °C, then 50 cycles at 95 °C for 15s and 60 °C for 1 min.

Assessment of microRNA gene expression was performed using the comparative Ct method [22]. The geometric mean of Ct values of selected endogenous controls (RNU58A and RNU38B) was used as a normalization factor [23] to normalize microRNA gene expression. Selection and validation of endogenous controls for microRNA expression studies in whole peripheral blood samples affected by pregnancy-related complications have already been described in our previous study [24]. In brief, expression of 20 candidate endogenous controls (HY3, RNU6B, RNU19, RNU24, RNU38B, RNU43, RNU44, RNU48, RNU49, RNU58A, RNU58B, RNU66, RPL21, U6 snRNA, U18, U47, U54, U75, Z30 and cel-miR-39) was investigated using NormFinder [25]. RNU58A and RNU38B were identified as the most stable small nucleolar RNAs (ncRNA) and equally expressed between patients with normal and abnormal courses of gestation. Therefore, these ncRNAs were selected as the most suitable endogenous controls for normalization of microRNA qPCR expression studies performed on whole peripheral blood samples affected by pregnancy-related complications.

### 2.4. Statistical Analysis

With regard to non-normal distribution of the data, microRNA gene expression was compared between normotensive pregnancies subsequently diagnosed to carry AGA, SGA, and FGR fetuses using the Kruskal–Wallis one-way analysis of variance. Afterward, a post-hoc test for the comparison among multiple groups and Benjamini–Hochberg correction for multiple comparisons was applied (Table 2) [26,27].

Box plots display the median, the 75th and 25th percentiles (the upper and lower limits of the boxes), the maximum and minimum values that are no more than 1.5 times the span of the interquartile range (the upper and lower whiskers), outliers (circles), and extremes (asterisks), respectively. Statistica software (version 9.0; StatSoft, Inc., Tulsa, OK, USA) was used to produce box plots.

Receivers operating characteristic (ROC) curve analyses state the areas under the curves (AUC), the best cut-off points related sensitivities, specificities, positive and negative likelihood ratios (LR+, LR-), sensitivities at 10.0% false positive rate (FPR), respectively (MedCalc Software bvba, Ostend, Belgium). To select the optimal microRNA combinations, logistic regression with subsequent ROC curve analyses were applied (MedCalc Software bvba, Ostend, Belgium).

## 3. Results

Expression profiles of cardiovascular disease-associated microRNAs in the whole peripheral venous blood were compared during the first trimester of gestation between women with appropriate-for-gestational-age (AGA) fetuses, small-for-gestational-age (SGA) fetuses, and fetal growth-restricted (FGR) fetuses. Results that reached a statistical significance after Benjamini–Hochberg correction for multiple comparisons are discussed below in detail (Appendix A). Statistical non-significant data gained after the application of Benjamini–Hochberg correction are also displayed (Appendix A) but not discussed further.

### 3.1. Early Dysregulation of Cardiovascular Disease-Associated MicroRNAs in Women with SGA or FGR Fetuses

Increased expression of miR-16-5p (*p* = 0.012^∗^, *p* = 0.029^∗^), miR-20a-5p (*p* < 0.001^∗∗∗^, *p* = 0.020^∗^), miR-146a-5p (*p* < 0.001^∗∗∗^, *p* < 0.001^∗∗∗^), miR-155-5p (*p* < 0.001^∗∗∗^, *p* = 0.004^∗∗^), miR-181a-5p (*p* < 0.001^∗∗∗^, *p* < 0.001^∗∗∗^), and miR-195-5p (*p* = 0.007^∗^, *p* = 0.022^∗^) was detected during the first trimester of gestation in SGA or FGR pregnancies. Upregulation of miR-1-3p (*p* < 0.001^∗∗∗^), miR-20b-5p (*p* < 0.001^∗∗∗^), miR-126-3p (*p* = 0.003^∗^), miR-130b-3p (*p* = 0.014^∗^), and miR-499a-5p (*p* = 0.009^∗^) was observed just in SGA pregnancies. In addition, mir-145-5p (*p* < 0.001^∗∗∗^), miR-342-3p (*p* = 0.004^∗^), and miR-574-3p (*p* = 0.002^∗∗^) represented unique upregulated early microRNA biomarkers in FGR pregnancies (Appendix A).

Apart from exceptions mostly very suitable sensitivities at 10.0% FPR for miR-16-5p (13.51%, 24.39%), miR-20a-5p (59.46%, 30.49%), miR-146a-5p (64.86%, 37.80%), miR-155-5p (8.11%, 1.20%), miR-181a-5p (32.43%, 30.49%), and miR-195-5p (10.81%, 19.51%) were detected in SGA or FGR pregnancies. Furthermore, miR-1-3p (56.76%), miR-20b-5p (21.62%), miR-126-3p (21.62%), miR-130b-3p (18.92%), and miR-499a-5p (21.62%) showed moderate to very suitable sensitivities at 10.0% FPR to distinguish between normal and SGA pregnancies. Upregulation of mir-145-5p (18.29%), miR-342-3p (20.73%), and miR-574-3p (26.83%) was also present in early stages of gestation in a proportion of FGR pregnancies at 10.0% FPR (Appendix A).

### 3.2. The Very Suitable Accuracy of First Trimester Combined MicroRNA Screening to Differentiate between FGR and Normal Pregnancies

The combined screening of seven microRNA biomarkers (miR-16-5p, miR-20a-5p, miR-145-5p, miR-146a-5p, miR-181a-5p, miR-342-3p, and miR-574-3p (AUC 0.725 *p* < 0.001, 42.68% sensitivity, 95.0% specificity, cut off >0.6578) revealed that at 10.0% FPR 42.68% FGR pregnancies had aberrant microRNA expression profile in early stages of gestation (Figure 1).

In our group of FGR pregnancies, the first-trimester screening detected 23 out of 82 pregnancies (28.05%) at risk of the development of PE and/or FGR using the criteria of the Fetal Medicine Foundation [10]. Prophylactic use of ASA was subsequently administered in total to 21 out of 82 women (25.61%). ASA was indicated as prevention of PE/FGR development on the basis of the positive results of the first-trimester screening for PE/FGR (*n* = 19) or with regard to the history of PE or FGR in anamnesis (*n* = 2). In four patients with the positive results of the first-trimester screening for PE or FGR, the usage of ASA was contraindicated.

The comparison of the predictive results of the routine first-trimester screening for PE or FGR based on the criteria of the Fetal Medicine Foundation [10] and the first-trimester screening for FGR using a panel of seven cardiovascular disease-associated microRNAs only revealed that the detection rate of FGR in our group of pregnancies increased 1.52-fold (42.68% vs. 28.05%).

### 3.3. The High Accuracy of First Trimester Combined MicroRNA Screening to Differentiate between SGA and Normal Pregnancies

The combined screening of four microRNA biomarkers (miR-1-3p, miR-20a-5p, miR-146a-5p, and miR-181a-5p) was able to detect at 10.0% FPR in early stages of gestation aberrant microRNA expression profile in 75.68% SGA pregnancies (AUC 0.868, *p* < 0.001, 75.68% sensitivity, 90.0% specificity, cut off >0.3664) (Figure 2).

In our group of SGA pregnancies, the first-trimester screening detected 6 out of 37 pregnancies (16.22%) at risk of the development of PE and/or FGR using the criteria of the Fetal Medicine Foundation [10]. Prophylactic use of ASA was finally administered to only 2 out of 37 women (5.40%) on the basis of the positive results of the first-trimester screening for PE or FGR.

The comparison of the predictive results of the routine first-trimester screening for PE or FGR based on the criteria of the Fetal Medicine Foundation [10] and the first-trimester screening for SGA using a panel of four cardiovascular disease-associated microRNAs only revealed that in our group of pregnancies the detection rate of SGA increased 4.67-fold (75.68% vs. 16.22%).

## 4. Discussion

Gene expression of 29 previously selected cardiovascular disease-associated microRNAs was compared in the whole peripheral venous blood during the early stages of gestation (within 10 to 13 weeks) between pregnancies carrying AGA, SGA, and FGR fetuses. Upregulation of miR-16-5p, miR-20a-5p, miR-146a-5p, miR-155-5p, miR-181a-5p, and miR-195-5p was detected during the first trimester of gestation in SGA or FGR pregnancies. In addition, upregulation of miR-1-3p, miR-20b-5p, miR-126-3p, miR-130b-3p, and miR-499a-5p was observed in SGA pregnancies only. Parallel, miR-145-5p, miR-342-3p, and miR-574-3p represented other upregulated early microRNA biomarkers in FGR pregnancies.

To our knowledge, no data on the prediction of SGA or FGR using microRNA profiling in maternal whole peripheral blood cell lysates during the early stages of gestation were reported. Parallel, few studies were dedicated to the assessment of the predictive potential of early circulating microRNA biomarkers for the development of FGR or SGA, which differs from biomarkers explored under the current study.

Kim et al. validated two microRNAs (miR-374a-5p and let-7d-5p) as potential plasma biomarkers for the early prediction of SGA. Increased expression of miR-374a-5p and let-7d-5p were found in plasma samples at 12–14^+6^ weeks of gestation derived from pregnancies delivering SGA newborns [28].

The study of Pei et al. reported a correlation of plasma miR-590-3p expression levels with FGR risk, but in middle (20th and 21st weeks of gestation) and late pregnancies (33th and 34th weeks of gestation) only, not in early stages of gestation (10th and 11th weeks of gestation). Increased plasma miR-590-3p levels were observed to be associated with an increased FGR risk via interaction with genes for vascular endothelial growth factor (VEGF), placental growth factor (PIGF), and matrix metalloproteinase 9 (MMP9), whose expression levels were significantly decreased [29].

Our group previously demonstrated that maternal plasma exosomal profiling of selected C19MC microRNAs revealed a novel microRNA biomarker (miR-520a-5p) that was down-regulated in early stages of gestation in pregnancies subsequently diagnosed to carry FGR fetuses [30]. Unfortunately, no circulating C19MC microRNA biomarkers were identified to predict the occurrence of FGR if maternal plasma samples were screened during the first trimester of gestation [31].

Recently, we have observed upregulation of miR-1-3p in a significant proportion of patients with chronic hypertension (51.72% at 10.0% FPR) [30]. The current study revealed that miR-1-3p expression is also dysregulated during the early stages of gestation in more than half of pregnancies carrying SGA fetuses (56.76% at 10.0% FPR).

Parallel, not long ago, we found out upregulation of miR-20a-5p and miR-146a-5p during early stages of gestation in women with chronic hypertension (44.83% and 65.52% at 10.0% FPR) and normotensive women with later occurrence of PE (33.33% and 42.42% at 10.0% FPR) [32]. At present, it is evident that upregulation of miR-20a-5p and miR-146a-5p also appears during the first trimester of gestation in a substantial proportion of pregnancies carrying SGA (59.46% and 64.86% at 10.0% FPR) or FGR fetuses (30.49% and 37.80% at 10.0% FPR).

Alongside, it is evident that miR-181a-5p upregulation may be detected in early stages of gestation not only in normotensive women subsequently developing hypertensive pregnancy-related complications such as GH (22.89% at 10.0% FPR) or PE (40.91% at 10.0% FPR) [32] but as well as in pregnancies affected with SGA (32.43% at 10.0% FPR) or FGR (30.49% at 10.0% FPR).

At the same time, upregulation of miR-145-5p and miR-574-3p is present in the early stages of gestation in a smaller proportion of normotensive pregnancies with subsequent onset of PE (21.21% and 27.27% at 10.0% FPR) [32] and those ones later diagnosed with FGR (18.29% and 26.83% at 10.0% FPR).

First-trimester screening in the whole peripheral venous blood based on the assessment of gene expression of these four selected cardiovascular disease-associated microRNAs (miR-1-3p, miR-20a-5p, miR-146a-5p, and miR-181a-5p) only led to the undeniable increase in the detection rate of SGA pregnancies when compared with the routine first-trimester screening for PE and/or FGR based on the criteria of the Fetal Medicine Foundation [11]. The detection rate of SGA pregnancies increased using this molecular biology approach 4.67-fold (75.68% vs. 16.22%). The combination of four upregulated microRNA biomarkers (miR-1-3p, miR-20a-5p, miR-146a-5p, and miR-181a-5p) seems to be superior over other microRNA combinations or the performance of individual microRNA biomarkers since it is able to reveal during early stages of gestation, majority of pregnancies carrying SGA fetuses (75.68% at 10.0% FPR).

Similarly, the detection rate of FGR pregnancies was increased 1.52-fold (42.68% vs. 28.05%) when compared with the results of the risk analysis using the criteria of the Fetal Medicine Foundation [11]. Nevertheless, in case of FGR pregnancies, the combination of seven microRNA biomarkers (miR-16-5p, miR-20a-5p, miR-145-5p, miR-146a-5p, miR-181a-5p, miR-342-3p, and miR-574-3p) was needed. The combination of these 7 selected microRNA biomarkers showed the best performance over other combinations of microRNA biomarkers or individual microRNA biomarkers and revealed at 10.0% FPR at early stages of gestation 42.68% of pregnancies diagnosed during the third trimester of gestation with FGR without PE.

Existing data suggest that microRNAs play a role in the pathogenesis of cardiovascular and cerebrovascular diseases. We focused mainly on those microRNAs playing a role in pathogenesis of obesity (miR-103a-3p, miR-181a-5p, miR-342-3p) [33,34,35,36,37], dyslipidaemia (miR-1-3p, miR-21-5p, miR-146a-5p, miR-155-5p) [38,39,40,41,42,43,44,45,46,47,48,49,50,51,52,53,54,55], hypertension (miR-21-5p, miR-103a-3p, miR-143-3p, miR-145-5p, miR-181a-5p) [56,57,58,59,60,61,62], vascular inflammation (miR-29a-3p, miR-126-3p, miR-146a-5p, miR-155-5p, miR-195-5p, miR-210-3p, miR-221-3p) [63,64,65], insulin resistance and diabetes (miR-20b-5p, miR-21-5p, miR-24-3p, miR-26a-5p, miR-29a-3p, miR-103a-3p, miR-126-3p, miR-133a-3p, miR-181a-5p) [47,66], atherosclerosis (miR-21-5p, miR-126-3p, miR-143-3p, miR-145-5p, miR-155-5p) [64,67,68,69,70,71,72], angiogenesis (miR-16-5p, miR-17-5p, miR-20a-5p, miR-21-5p, miR-92a-3p, miR-100-5p, miR-126-3p, miR-210-3p, miR-221-3p) [73,74,75], coronary artery disease (miR-1-3p, miR-17-5p, miR-20a-5p, miR-21-5p, miR-92a-3p, miR-126-3p, miR-133a-3p, miR-143-3p, miR-145-5p, miR-155-5p, miR-181a-5p, miR-195-5p, miR-221-3p) [60,65,73,76,77,78,79], myocardial infarction, heart ischemia and heart failure (miR-1-3p, miR-16-5p, miR-17-5p, miR-20b-5p, miR-21-5p, miR-23a-3p, miR-24-3p, miR-26a-5p, miR-29a-3p, miR-92a-3p, miR-100-5p, miR-125b-5p, miR-126-3p, miR-103a-3p, miR-133a-3p, miR-181a-5p, miR-195-5p, miR-199a-5p, miR-210-3p, miR-499a-5p) [57,80,81,82,83,84,85,86,87,88,89,90,91,92,93,94,95,96,97,98,99,100,101,102,103,104,105,106,107,108,109,110,111,112], stroke (miR-17-5p, miR-125b-5p, miR-146a-5p, miR-181a-5p) [113,114,115,116,117], and pulmonary hypertension (miR-20a-5p, miR-103a-3p) [118,119].

Multiple predicted targets of appropriate dysregulated microRNAs have already been reported to be involved in particular human biological pathways related to abnormal pregnancy outcomes and their associated cardiovascular risk. These biological pathways include, for example, apoptosis pathway, an inflammatory response pathway, senescence and autophagy pathways [120], insulin signaling pathway, type 1 diabetes mellitus pathway, and type 2 diabetes mellitus pathway [121].

In addition, interactions between microRNAs dysregulated in the whole peripheral blood of preterm-born children and specific genes related to human disease ontologies, such as hypertension, congenital heart defects, patent foramen ovale, patent ductus arteriosus, and heart valve disease, were also demonstrated in our previous study [122].

Furthermore, we previously reported an extensive file of predicted targets of all microRNAs aberrantly expressed in the whole peripheral blood of children descending from GDM complicated pregnancies. A large group of these genes is involved in ontologies of human diseases (obesity, hypertension, glucose intolerance, lipid metabolism disease, type 2 diabetes mellitus, heart septal defects and heart valve disease, heart disease, heart failure, venous insufficiency, and pulmonary embolism) [123].

## 5. Conclusions

Large-scale follow-up studies have to be performed to verify the diagnostical potential of early cardiovascular disease-associated microRNA biomarkers to predict the subsequent occurrence of SGA (miR-1-3p, miR-20a-5p, miR-20b-5p, miR-126-3p, miR-130b-3p, miR-146a-5p, miR-181a-5p, miR-499a-5p) or FGR (miR-16-5p, miR-20a-5p, miR-145-5p, miR-146a-5p, miR-181a-5p, miR-195-5p, miR-342-3p, and miR-574-3p). However, the verification of the data resulting from the current study through the mediation of large-scale follow-up studies is highly demanding. Tens of thousands of pregnancies have to be followed from the first trimester of gestation until the delivery to acquire a sufficient amount of SGA and FGR pregnancies.

However, it seems probable that microRNAs represent promising biomarkers with very suitable predictive potential for SGA or FGR and might be implemented into current routine first-trimester screening programs to predict the later occurrence of SGA or FGR. The comparison of the predictive results of the routine first-trimester screening based on the criteria of the Fetal Medicine Foundation [10] and the first-trimester screening using panels of selected cardiovascular disease-associated microRNAs only revealed that the detection rates of SGA and FGR increased 4.67-fold and 1.52-fold. The combination of four microRNA biomarkers (miR-1-3p, miR-20a-5p, miR-146a-5p, and miR-181a-5p) seems to be optimal to predict the later occurrence of SGA without PE. The combination of seven microRNA biomarkers (miR-16-5p, miR-20a-5p, miR-145-5p, miR-146a-5p, miR-181a-5p, miR-342-3p, and miR-574-3p) is needed to identify in early stages of gestation subsequent occurrence of FGR without PE.

## 6. Patents

National patent application—Industrial Property Office, Prague, Czech Republic (Patent No. PV 2021-562).

## Figures and Tables

**Figure 1 biomedicines-10-00718-f001:**
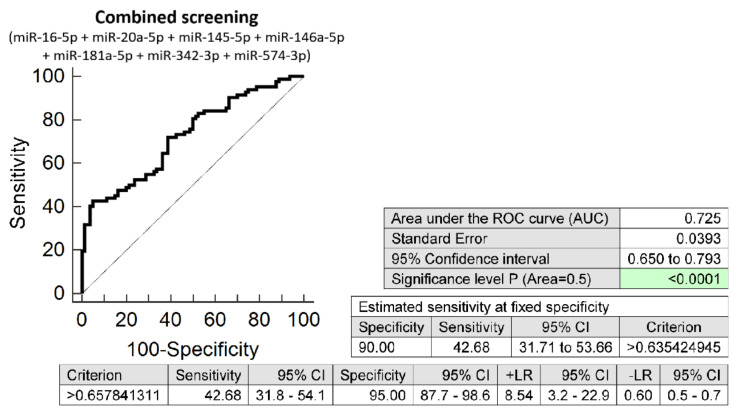
ROC analysis—the combination of microRNA biomarkers. The combination of miR-16-5p, miR-20a-5p, miR-145-5p, miR-146a-5p, miR-181a-5p, miR-342-3p, and miR-574-3p showed that at 10.0% FPR 42.68% FGR pregnancies had aberrant microRNA expression profile in the whole peripheral venous blood during the first trimester of gestation.

**Figure 2 biomedicines-10-00718-f002:**
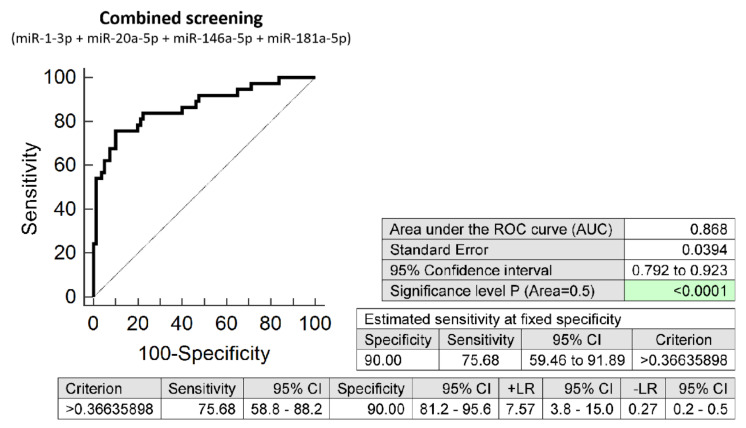
ROC analysis—the combination of microRNA biomarkers. The combination of miR-1-3p, miR-20a-5p, miR-146a-5p, and miR-181a-5p showed that at 10.0% FPR, 75.68% of SGA pregnancies had aberrant microRNA expression profile in the whole peripheral venous blood during the first trimester of gestation.

**Table 1 biomedicines-10-00718-t001:** Clinical characteristics of the controls and complicated pregnancies.

	Normal Pregnancies (*n* = 80)	SGA w/o PE (*n* = 37)	FGR w/o PE (*n* = 82)	*p*^1^-Value	*p*^2^-Value
** *Maternal characteristics* **					
Maternal age (years)	32 (25–42)	32 (23–43)	34 (22–44)	0.817	0.014
Advanced maternal age (≥35 years old)	20 (25%)	11 (29.73%)	37 (45.12%)	0.712	0.157
Caucasian ethnic group	80 (100%)	37 (100%)	82 (100%)	-	-
Prepregnancy BMI (kg/m^2^)	21.28 (17.16–29.76)	20.67 (17.01–38.86)	21.57 (15.92–37.39)	0.513	0.450
Diabetes mellitus (T1DM, T2DM)	0 (0%)	1 (2.70%)	2 (2.44%)	-	-
Autoimmune diseases (SLE/APS/RA)	0 (0%)	0 (0%)	2 (2.44%)	-	-
Chronic hypertension	0 (0%)	1 (2.70%)	7 (8.54%)	-	-
Parity					
Nulliparous	40 (50.0%)	22 (59.46%)	53 (64.63%)	0.340	0.060
Parous	40 (50.0%)	15 (40.54%)	29 (35.37%)		
ART (IVF/ICSI/other)	2 (2.5%)	6 (16.22%)	16 (19.51%)	0.006	<0.001
Smoking during pregnancy	2 (2.5%)	1 (2.70%)	2 (2.44%)	0.948	0.980
** *Pregnancy details (First trimester of gestation)* **					
Gestational age at sampling (weeks)	10.29 (9.57–13.71)	10.51 (9.86–13.0)	10.43 (9.86–13.43)	0.105	0.132
MAP (mmHg)	88.75 (67.67–103.83)	88.37 (76.83–105.58)	91.50 (71.25–127.0)	0.981	0.106
MAP (MoM)	1.05 (0.84–1.25)	1.05 (0.92–1.25)	1.08 (0.85–1.48)	0.911	0.128
Mean UtA-PI	1.39 (0.56–2.43)	1.53 (0.81–2.48)	1.50 (0.77–2.44)	0.196	0.045
Mean UtA-PI (MoM)	0.90 (0.37–1.55)	0.97 (0.51–1.59)	0.97 (0.48–1.56)	0.265	0.112
PIGF serum levels (pg/mL)	27.1 (8.1–137.0)	26.4 (10.6–67.0)	23.1 (6.20–88.1)	0.490	0.001
PIGF serum levels (MoM)	1.04 (0.38–2.61)	0.91 (0.44–1.70)	0.90 (0.30–1.99)	0.097	0.001
PAPP-A serum levels (IU/L)	1.49 (0.47–15.69)	1.68 (0.21–28.82)	1.06 (0.13–7.00)	0.753	<0.001
PAPP-A serum levels (MoM)	1.17 (0.37–3.18)	0.99 (0.21–4.19)	0.76 (0.12–3.10)	0.281	<0.001
Screen-positive for PE and/or FGR by FMF algorithm	0 (0%)	6 (16.22%)	23 (28.05%)	-	-
Aspirin intake during pregnancy	0 (0%)	2 (5.40%)	21 (25.61%)	-	-
** *Pregnancy details (At delivery)* **					
Systolic blood pressure (mmHg)	122 (100-141)	120 (100–145)	120 (86–159)	0.598	0.724
Diastolic blood pressure (mmHg)	76.5 (60.0–90.0)	80.0 (55.0–103.0)	80.0 (57.0–108.0)	0.469	0.118
Gestational age at delivery (weeks)	40.07 (37.57–42.0)	39.0 (33.00–40.29)	37.43 (27.71–41.43)	<0.001	<0.001
Delivery at gestational age <37 weeks	0 (0%)	4 (10.81%)	29 (35.36%)	-	-
BMI (kg/m^2^)	26.66 (21.71–34.82)	25.86 (20.57–39.56)	26.67 (17.65–41.68)	0.338	1.000
Weight gain during pregnancy (kg)	14 (3–25)	12 (1–24)	11 (1–25)	0.007	<0.001
Fetal birth weight (grams)	3470 (2920–4240)	2690 (1870–3360)	2335 (540–3010)	<0.001	<0.001
Fetal sex					
Boy	40 (50.0%)	17 (45.95%)	39 (47.56%)	0.683	0.756
Girl	40 (50.0%)	20 (54.05%)	43 (52.44%)		
Mode of delivery					
Vaginal	69 (86.25%)	19 (51.35%)	28 (34.15%)	<0.001	<0.001
CS	11 (13.75%)	18 (48.65%)	54 (65.85%)		

Continuous variables, compared using the Mann–Whitney test, are presented as median (range). Categorical variables, presented as number (percent), were compared using chi-squared test. *p*-value ^1,2^; the comparison among normal pregnancies and SGA or FGR, respectively. SGA, small-for-gestational-age; FGR, fetal growth restriction; BMI, body mass index; T1DM, type 1 diabetes mellitus; T2DM, type 2 diabetes mellitus; SLE, systemic lupus erythematosus; APS, antiphospholipid syndrome; RA, rheumatoid arthritis; ART, assisted reproductive technology; IVF, in vitro fertilization; ICSI, intracytoplasmic sperm injection; MAP, mean arterial pressure; UtA-PI, uterine artery pulsatility index; PIGF, placental growth factor; PAPP-A, pregnancy-associated plasma protein A; FMF, Fetal Medicine Foundation; CS, Cesarean section.

**Table 2 biomedicines-10-00718-t002:** Benjamini–Hochberg correction for multiple comparisons: comparison of microRNA gene expression between various groups of pregnancies (AGA pregnancies vs. FGR pregnancies vs. SGA pregnancies).

K	i	Alpha = 0.05	Alpha = 0.01	Alpha = 0.001
**3**		**0.05**	**0.01**	**0.001**
	**1**	0.017	0.003	0.000
	**2**	0.033	0.007	0.001
	**3**	0.050	0.010	0.001

## Data Availability

The data presented in this study are available on request from the corresponding author. The data are not publicly available due to rights reserved by funding supporters.

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
