# Peer review of "First-Trimester Screening for Fetal Growth Restriction and Small-for-Gestational-Age Pregnancies without Preeclampsia Using Cardiovascular Disease-Associated MicroRNA Biomarkers"

_biomedicines, 2022, doi:10.3390/biomedicines10030718_

Round 1

Reviewer 1 Report

The authors showed the combination of 7 microRNA biomarkers (miR-16-5r, miR-20a-5r, miR-145-5r, miR-146a-5r, miR-181a-5r, miR-342-3r, and miR-574- 3p) was able to identify 42.68% of FGR pregnancies at 10.0% FPR in the early stages of gestation. Indeed, CVD-associated microRNAs represent promising early biomarkers with good predictive potential for SGA or FGR without PE to be implemented into the routine screening programs. The relevance of the work is beyond doubt. However, the authors need to clarify the details of the experiment.

Some guidance is provided below:

  1. The authors collected biological material from 11/2012 to 3/2020. How was the long-term storage of the biological material performed? Were buffers used to stabilize the RNA during the storage of the material?
  2. Describe or explain the method used to obtain lysates whole peripheral blood cells? Explain the advantages of the work/justify the use of whole peripheral blood cell lysates instead of serum/plasma for predictive value analysis.
  3. How was the quality, purity, and concentration of total RNA assessed? Indicate the concentration of total RNA used for subsequent analysis. Include information on materials and methods.
  4. Add information about the amplification protocol.
  5. Several known controls exist (18S rRNAs, GAPDH, ACTB, sn/snoRNA, miRNA, RNU48, RNU44, U18, and U47). Explain why the following controls were used - RNU58A, RNU38B.
  6. The authors used a microRNA profile associated with cardiovascular disease. However, there is limited information on the role of differentially expressed miRNAs in the dysregulation of genes and signaling pathways in cardiovascular disease. We recommend expanding the discussion chapter to describe the possible regulatory relationships that contribute to abnormal pregnancy outcomes.

Reviewer 2 Report

Powerful statistics.

The study is analyzing an important number of cases during a long period of time in a reference national center and the number of patients is important providing a powerful statistics interpretation.

It is well documented;.

The aim of the study is clear and precise definite, the design is proper and the statistic analysis is clear.

There is no plagiarism detected, the bibliography is properly chosen.

I could not detect any self citations.

Author Response

Thank you very much for a positive evaluation of our research. We appreciate it.